# Resilience for Undergraduate Students: Development and Evaluation of a Theory-Driven, Evidence-Based and Learner Centered Digital Resilience Skills Enhancement (RISE) Program

**DOI:** 10.3390/ijerph191912729

**Published:** 2022-10-05

**Authors:** Wei How Darryl Ang, Shefaly Shorey, Zhongjia James Zheng, Wai Hung Daniel Ng, Emmanuel Chih-Wei Chen, Lubna Binte Iskhandar Shah, Han Shi Jocelyn Chew, Ying Lau

**Affiliations:** Alice Lee Centre for Nursing Studies, Yong Loo Lin School of Medicine, National University of Singapore, Singapore 119077, Singapore

**Keywords:** resilience, undergraduate students, mixed methods, feasibility study

## Abstract

Protective factors that build students’ resilience are known. A six-week digital resilience training program was developed on the basis of theory, evidence, and contextual information. The feasibility study sought to evaluate the acceptability, appropriateness, demand, implementation, and limited efficacy of a digital resilience skills enhancement program for undergraduate students. A single group, pre-test, post-test, concurrent mixed methods design among 10 undergraduate students was conducted in one university in Singapore. The content analysis concluded that students accepted and perceived the digital resilience skills enhancement program as appropriate. Students also proposed several improvements, such as the initiation of the program and revisions to the content. The Wilcoxon signed-rank test found significant improvements in resilience (*p* = 0.02) and meta-cognitive self-regulation (*p* = 0.01) scores with medium (d = 0.79, 95% CI: −0.15 to 1.74) and very large effect sizes (d = 1.31, 95% CI: 0.30–2.33), respectively. Students found the digital resilience program appropriate and were able to apply their newly acquired skills to promote their resilience and learning. Although, several improvements are proposed to enhance the rigor of the digital resilience program, the findings of this study suggests that digital resilience programs are important for students’ well-being.

## 1. Introduction

The concept of resilience has shifted since its inception. The initial classification of resilience has been associated with individual traits comprising intrinsic factors, such as coping skills and being positive [1]. With the expansion of the resilience research, extrinsic factors such as relationships with others and resources within the environment positively influencing one’s resilience are increasingly acknowledged [1,2]. Contemporary resilience scholars now view resilience as a process, where protective factors (e.g., coping or resources) play an unequivocal role in inoculating an individual against psychological distresses [2,3].

In line with the contemporary theoretical stance on resilience, this study adopts the definition by Van Breda [4] that is “the multilevel processes that systems engage in to obtain better-than-expected outcomes in the face or wake of adversity” [4]. According to Van Breda [4], multilevel processes refer to various mechanisms that may occur at the individual (e.g., positive mindset) or environmental level (e.g., community resources) in which resilience is fostered. Systems refer to the ‘unit’ that is engaging in these processes and can be scaled from the cellular (e.g., physiological response to stress) to the societal level (e.g., interactions with other individuals in the community) [4].

A better-than-expected outcome refers to a contextualized range of outcomes (e.g., resistance) across a population [5,6]. This meant that a considerably good outcome is determined by the individual within a certain context. This allows the individual to determine what constitutes a positive outcome in the wake of adversity [4]. This proposes that resilience can be developed and enhanced.

In spite of the undisputable benefits of resilience in students’ higher education (HE) journey, a meta-analysis reports a global prevalence of 36% for low resilience among HE students [7]. Students with lower resilience are associated with poorer well-being manifested through high level of stress, drop out from school, and poor performances in coursework [8,9]. Higher levels of stress can result in negative impacts on students’ learning outcomes in several ways [10,11]. Studies have shown that students experiencing high levels of stress have reduced pleasure in studying [12] and impaired memory retrieval [13]. When students are unable to learn properly, they may experience greater stress and anxiety [10,11]. Considering that being resilient can have protective effects against stress and mental health [9], it highlights the direct and indirect roles that resilience plays in supporting students’ learning and stress outcomes.

In addition, the presence and persistent aftermath effects of the coronavirus disease 2019 (COVID-19) pandemic led to heightened mental health issues among HE students [14,15]. Given the magnitude of the potential downstream implications, a resilience training program must be developed to promote HE students’ mental well-being, learning outcomes, and academic success.

The interest in resilience training and its impacts is considerable. Two recent meta-analyses [16,17] positioned resilience training as a potential universal prevention strategy that should be incorporated into the formal university curriculum. Specifically, the review concludes that students’ resilience and mental well-being (e.g., depressive and stress symptoms) improve following resilience training [16]. Although resilience is described to be influenced by personal, relational, and environmental protective factors, these are not consistently incorporated into the existing resilience training. This calls for a closer examination of how resilience training can incorporate personal, relational, and environmental factors.

With greater advancements in the technological infrastructure and need for remote learning due to the COVID-19 pandemic, institutes of higher learning (IHL) have transited to the digital platform as a medium of instruction [18,19]. Although the use of an asynchronous online learning platform potentially addresses the cost and scalability issues, drawbacks include the absence of institutional support and poor engagements [20]. Substantial thought about the curriculum design with a combination of learner-centered activities (e.g., quizzes and forums) is needed to engage learners [21].

On the contrary, blended learning considers the use of synchronous (e.g., traditional lectures) and asynchronous engagement tools (e.g., forum discussions) and integrates the merits of face-to-face and online learning [22]. However, it is resource-intensive and more challenging to implement compared with a solely asynchronous online learning approach.

Considering the scarcity of resources, preliminary evaluations using a feasibility study design are useful to ensure that a newly curated digital resilience training program is appropriate and sustainable [23]. The guidelines [16] for developing a feasibility study comprise the following domains: (1) acceptability (i.e., appropriateness or suitability), (2) demand (i.e., actual use), (3) implementation (i.e., facilitators or barriers), (4) practicability (i.e., ability to carry out activities), (4) adaptability (i.e., work in a new platform), (5) integration (i.e., fit in the infrastructure), (6) expansion (i.e., ability to fit in the organization), and (7) limited efficacy testing (i.e., intended effects).

To the authors’ knowledge, few studies examine the acceptability, demand, practicability, and treatment dose effect. In addition, resilience interventions thus far are developed and evaluated in other contexts and countries. Given that the environment plays an important part in developing resilience; developing and evaluating a context-specific resilience intervention are important. Therefore, this study seeks to develop and evaluate a digital resilience skills enhancement (RISE) program for undergraduate students using a feasibility study design.

## 2. Methods

### 2.1. Design

A single-arm, pre-test, post-test feasibility study was conducted from October to November 2021. A concurrent mixed-methods approach was used to assess the acceptability, appropriateness, demand and actual use, and limited-efficacy testing of the digital RISE program [23]. The qualitative component explored students’ acceptability, appropriateness, demand, and actual use, whereas the quantitative approach was used to examine the usage patterns and limited efficacy testing on resilience, social support, and learning scores. This study was prospectively registered on the clinical trials register (NCT05072340).

### 2.2. Participants and Setting

The digital RISE program was offered to all full-time undergraduate students in one university in Singapore. Advertisement posters were disseminated through the university’s online learning and social media platforms. A convenience sampling approach was used to recruit students. Participants were eligible if they were a full-time undergraduate, above 18 years old, could comprehend the English language, and had a device that could connect to the Internet. Students with any self-reported medical history of any mental health conditions or that received similar psychosocial interventions were excluded from this study.

### 2.3. Development of the Digital RISE Program

The digital RISE program was developed using a three-step approach for the development of complex interventions with reference to Medical Research Council guidance [24]. First, a theoretical basis was formed to explain the mechanism of the intervention. Second, evidence was gathered to deepen the understanding of the effects of resilience training. Finally, to ensure that the intervention is contextually relevant and user-centered, key stakeholders’ perception and preferences for receiving resilience training were solicited. Figure 1 presents the schematic of the development of the digital RISE program.

#### 2.3.1. Theory

The society-to-cell resilience theory [25] and the transactional model of stress and coping [26] were used to provide the theoretical basis. Resilience theory proposes that resilience may be enhanced through several protective factors [25]. The protective factors include personal (e.g., coping abilities), relational (e.g., presence of secure social networks), and environmental aspects (e.g., availability of resources).

Given that resilience may be developed after experiencing adversity, the transactional model of stress and coping [26] was used to map the causal pathway of how the RISE program may work. With these two as bases, this study postulated that, when a resilient individual experiences stress, they can rely on resilience protective factors to overcome the challenges.

#### 2.3.2. Literature Evidence

Evidence was gathered by conducting three systematic reviews [16,17,27]. First, the meta-ethnography of qualitative studies concluded that a higher students’ resilience is influenced by personal, relational, and environmental factors [27]. Beyond the known personal and relational factors [27,28], it highlights the importance of the IHLs in shaping students’ resilience, suggesting that resilience skills should encompass the introduction of school-based services [19].

Second, the meta-analysis [16] further highlights the role of relationships in enhancing students’ resilience and proposes that resilience training should comprise skills that build social competencies. Two systematic reviews were conducted to identify the features of resilience programs that could best promote resilience [16,17]. The findings proposed that the resilience program should comprise of lectures and discussions delivered over the Internet and adopt a flexible schedule [16,17].

#### 2.3.3. Contextual Information

As part of the development process, a qualitative study was conducted to ensure contextual relevance to the RISE program [28]. The study gathered students’ suggestions and preferences for receiving resilience training and provided insight into the content development of the RISE program. From a design perspective, students proposed the use of blended learning pedagogy and short videos. In addition, students recommended using alternative forms of information delivery, such as narratives from resilient individuals and utilizing reflective practices. They also proposed that the RISE program should be introduced during the school term, right before major assessments, to allow students to apply their newly acquired skills [22].

### 2.4. Digital RISE Program

The information gathered from the theories, evidence (i.e., systematic reviews), and contextual information (e.g., qualitative study) was consolidated and mapped. The digital RISE program was designed as six weekly sessions that were hosted on the university’s online learning platform (Figure 2) and made available through a mobile application. The six sessions included: (1) introduction to resilience and embracing change, (2) coping strategies, (3) creating positivity, (4) shifting mindsets, (5) building social competency, and (6) preparing for the future. The details of each session are described in Table 1.

The digital RISE program adopted the blended learning framework [29]. In each week, students were provided access to review the online resources (e.g., reading materials and videos) at their own pace. Online videos comprised traditional lectures or narratives in the form of interviews with invited content experts. In light of the COVID-19 pandemic and numerous competing priorities, physical sessions were replicated in a virtual setting and were held remotely via virtual face-to-face discussions through Zoom (Weeks 1, 5, and 6) and forums (Weeks 2, 3, and 4). At the end of each week, participants were provided with take home tasks and quizzes that aimed to consolidate and evaluate their learning outcomes. Weekly reminders were sent via the learning platform’s announcement function and emails. The total possible duration (inclusive of videos and virtual discussions) for the RISE program was 365.89 min.

### 2.5. Content Validation

Content validation was conducted to assess the quality and accuracy of the RISE program. The health-related website evaluation tool [30] comprised 36 items across 8 domains, namely (1) content, (2) accuracy, (3) author, (4) currency, (5) audience, (6) navigation, (7) external links, and (8) structure. The items were rated agree (2 points), disagree (1 point), or not applicable. The scores were summed and the quality was assessed based on the following cut-off scores: poor (<75%), adequate (75%), or excellent (at least 90%). A total of five content experts, specifically an educator, student support manager, mental health practitioner, executive coach, and researcher, were invited to validate the contents (Appendix A). The scores ranged from 94% to 100%, suggesting the excellence of the RISE program [30].

### 2.6. Data Collection

Students were recruited from September to October 2021. Students interested in the study expressed their interest through a link provided in the advertisement posters. The primary researcher approached the interested participants and provided them with more information. All clarifications were addressed prior to the obtained informed consent. Participants completed the pretest measures prior to the commencement of the RISE program in October 2021. Following the conclusion of the intervention, students were invited to the complete the post-test in November 2021. The survey comprised a sociodemographic form and three outcome measures (resilience, social support, and learning). The qualitative data were collected via email correspondence. Participants were provided with a list of open-ended questions and given time to respond. Participants were provided with a remuneration of 10 SGD for their participation.

### 2.7. Outcome Measures

The quantitative measures were collected using Qualtrics and comprised the following domains: (1) sociodemographic data, (2) resilience, (3) social support, and (4) learning. The sociodemographic, academic, and economic characteristics of the students were solicited. Open-ended questions were developed with reference to the guidelines of Bowen and Kreuter [16] for feasibility studies.

#### 2.7.1. Resilience

Resilience was measured using the Connor-Davidson Resilience scale that consists of 25 self-rated items (CD-RISC-25) [31]. The items were rated from “not true at all” (0) to “true nearly all the time” (4), with a higher score suggesting greater resilience. The CD-RISC has the best psychometric properties when compared with other measures of resilience [32], with a Cronbach’s alpha of 0.89 [31].

#### 2.7.2. Social Support

Social support was assessed using the Multidimensional Scale of Perceived Social Support (MSPSS) [33]. The MSPSS comprised 12 items and assessed students’ social support from three sources, namely family, friends, and significant others. The items were rated on a 7-point Likert scale from “very strongly disagree” (1) to “very strongly agree’ (7), with a higher greater score proposing that students have greater access to social support [33]. The MSPSS demonstrates good psychometric properties with a Cronbach’s alpha ranging from 0.84 to 0.92 [34,35].

#### 2.7.3. Learning

Motivated Strategies for Learning Questionnaire (MSLQ) [36] was used to examine the effect of the RISE program on students’ motivational orientations and learning outcomes. MSLQ consisted of 81 items across two domains: (1) a cognitive view of motivation and (2) learning strategies [30]. This study adopted three subscales, each from the motivation and learning strategies domains. The three subscales from the motivation domain were the control of learning beliefs, self-efficacy for learning performances, and test anxiety. The three subscales from the learning domain consist of meta-cognitive self-regulation, time and study environment, and effort regulation. The items were rated on a 7-point Likert scale, ranging from “not true at all of me” (1) to “very true of me” (7), with higher scores indicating higher motivation or the better use of learning resources [36]. MSLQ has sound psychometric properties [36,37].

#### 2.7.4. Participants’ Acceptance and Usage Patterns

Open-ended questions were curated to explore participants’ acceptability, perceived appropriateness, and actual usage of the intervention (Appendix A). The usage patterns comprised data such as participants’ access time, viewing frequency, duration, and completion rate. The collected data were used to address numerous aspects of feasibility testing and to corroborate participant’s acceptance, assess the demand, and review the implementation of RISE.

### 2.8. Data Analysis

#### 2.8.1. Quantitative Analysis

Stata version 17 [38] was used for the data analysis. All survey data were deidentified prior to analysis. Descriptive statistics, such as the median and interquartile ranges, were used to report continuous variables, whereas the frequency and percentage were used to present categorical variables. The Wilcoxon signed-ranked test was used to compare the median and interquartile ranges across two time points. Box plots were used for graphical representations. Effect sizes were used to measure the magnitude of the differences between the means at the pretest and post-test [39]. Effect sizes were computed as Cohen’s d with the following cut-offs: small (d = 0.2), medium (d = 0.5), large (d = 0.8), and very large (d =1.3) [39].

#### 2.8.2. Qualitative Analysis

A content analysis [40] was used to analyze the qualitative data. First, two researchers (DA and LY) independently reviewed the participants responses to the open-ended questions to increase their familiarity with the data. Second, the authors used different colored highlighters to perform manual coding independently. The codes were exported into a tabular format using Microsoft Excel. Third, the two sets of codebooks were brought together for comparison, and discussions were frequently held to resolve all discrepancies. Inter-rater agreements were calculated using Cohen’s Kappa, κ, with a value of 1 suggesting perfect agreement to −1 as the absence of agreement [41]. Values above 0.75 were considered excellent in agreement, while values between 0.60 and 0.75 were considered a good agreement [41]. The inter-rater agreement of the selected codes was 0.64, suggesting a good agreement. A third author (JC) was brought in for further deliberations. Finally, the categories were developed independently and compared for cohesiveness.

#### 2.8.3. Mixed Data Analysis

The concurrent mixed data analysis approach with seven steps [42] was undertaken to integrate the quantitative and qualitative data. The first step involved the display of quantitative and qualitative data. Second, the respective datasets were reduced. Third, the quantitative data (e.g., results of CD-RISC, MSPSS, and MSLQ) were qualitatively transformed into narratives. The content analysis of the qualitative data documented the frequencies of the categories provided by each student, which transformed the qualitative data into quantitative data. In steps four to six, both sets of data were consolidated, compared, and used to explain both sets of findings. Finally, the data were integrated into one coherent outcome.

### 2.9. Ethical Considerations

This study conformed to the Declaration of Helsinki. Ethical approval was obtained from the university’s institutional review board (NUS-IRB-2021-594) prior to the commencement of this study. Participants were informed of the study’s goals and were provided opportunities to clarify their doubts. Written informed consent was obtained from all participants.

## 3. Results

### 3.1. Participant Characteristics

A total of 43 students were assessed for eligibility. Fifteen students did not meet the inclusion criteria, and 18 students decided not to participate. An eventual sample of 10 undergraduate students enrolled in the RISE program, with a mean age of 21.5 years (SD = 1.96). The participants were largely ethnic Chinese (90%), female (60%), of Buddhist faith (40%), and single (100%). The demographic information of the students is detailed in Table 2. One participant dropped out and did not complete the program. Nine participants completed the post-intervention survey. Students’ narratives are presented in Table 3. 

### 3.2. Perceived Appropriateness

Regarding the content, all participants found the materials appropriate and applicable to their concerns. In addition, participants appreciated how the take-home tasks were able to consolidate and support their learning. The use of reflective practices was not well-received, and students proposed alternative forms of take-home tasks to stimulate their interest. Regarding the training duration, all participants felt that the duration of six weeks was suitable to deliver the program, although the duration of the training could be shortened. When asked to propose other forms of resilience-enhancing strategies, participants alluded to the importance of skills that fostered their academic coping. Finally, participants also emphasized the need for information relating to outlets for seeking professional support.

### 3.3. Implementation

The majority of the students verbalized that the training had positive impacts on their resilience, social support, and learning outcomes. In particular, students appreciated how they transferred these skills to their everyday lives. However, the RISE program started during the middle of the academic semester. Students proposed that the training could be initiated at the start of the semester as the current time frame was challenging because of numerous competing academic priorities.

### 3.4. Demand and Actual Use

The majority of the students were able to complete the videos (*n* = 8), quizzes (*n* = 9), and take-home tasks (*n* = 7) and attended at least two of the three virtual face-to-face discussions (*n* = 8). This led to a cumulative completion rate of 72% (Range: 60–100%, SD = 28.42), and their total engagement with the RISE program was 182.66 min (Range: 121–360 min, SD = 92.85). Consistent with the students’ narratives and academic timetables, their engagement with the materials was reduced by weeks 5 and 6 of the RISE program, which were also the final two weeks of the academic semester, wherein their workload was heavier.

### 3.5. Limited Efficacy Testing

The Wilcoxon signed-ranked test was used to compare the differences between the baseline and post-intervention (Table 4). The findings suggested that the RISE program was effective in improving students’ resilience (Z = −2.37, *p* = 0.01) (Figure 3), with a medium effect size (d = 0.79, 95% CI: −0.15 to 1.74). No significant differences were found for social support (Z = −1.67, *p* = 0.09) and its subscales (Appendix A). Regarding learning (Appendix A), significant differences were noted for meta-cognitive self-regulation (Z = −2.43, *p* = 0.01), with a very large effect (d = 1.31, 95% CI: 0.30–2.33). No changes were observed for the other MSLQ subscales.

### 3.6. Integration of Qualitative and Quantitative Data

The mixed data analysis [42] comprised qualitative and quantitative data (Appendix A). The qualitative and quantitative data were displayed, reduced, and transformed. Both sets of data were then brought together and compared. The main findings of the mixed-methods analysis concluded that students found the RISE program acceptable and appropriate and that it improved their resilience and meta-cognitive self-regulation. However, modifications were needed with regards to the initiation and contents of the RISE program.

## 4. Discussion

This mixed-methods feasibility study evaluated the theory-driven, evidence-based, and learner-centered digital resilience training program among undergraduate students. The qualitative results showed that students found the intervention acceptable and appropriate and that they were able to engage with the RISE program. The quantitative results found that students’ resilience and meta-cognitive self-regulation scores improved after the RISE program. In addition, the students’ completion rate was 72%. This figure was comparable to other reported resilience programs whose completion rates ranged from 55% [43] to 78% [44].

Although evaluating the efficacy of an intervention is not the goal of a feasibility study [23], this study found that resilience and meta-cognitive self-regulation scores improved significantly at the post-test. This study found that students’ resilience improved following the RISE program. This was similarly reported in two meta-analyses of resilience programs [16,17]. The positive findings could be attributed to several reasons.

One, the contents of the program were developed based on theories [25,26] where personal, relational, and environmental protective factors of resilience were introduced to improve students’ resilience. Based on the resilience [25] and stress-coping [26] theories, these factors may work in synergy to promote one’s ability to overcome adversities and thus improve their resilience and self-regulation. For instance, the RISE program provided skills that promoted students’ positivity and gratitude, and this was found to improve students’ learning attitudes by allowing them to see the value in and become motivated to exert effort towards learning [45,46]. By becoming motivated toward learning, it is postulated to have a buffering effect against anxiety [47,48].

Two, as part of the developmental process, a qualitative study ensured that the contents and features of the RISE program were cocreated with the target population [28]. With regards to the content, the use of a cocreation approach can ensure the contextual relevance of the intervention and thus ensure that the intervention is relevant to participants’ priorities and needs; this, in turn, improves the outcomes [49,50]. This finding was also supported in two studies evaluating resilience programs [51,52], where the contextual relevance of the resilience programs supported participants’ learning.

Three, with regards to the features, the RISE program was hosted on the university’s learning platform. With reference to the theory of online learning [51], placing a digital program in a platform that is familiar to the target audience can facilitate their participation. In addition, the specific features of the RISE program were developed based on a cocreation process [28], wherein the design features were tailored to students’ preferences, thus enabling their sustained participation.

Four, the RISE program was largely designed as a self-paced program, allowing learners to have flexibility and take ownership of their learning, which, in turn, created a learner-centered environment [53,54]. In addition, the combination of take-home tasks, quizzes, and virtual face-to-face or forum discussions were opportunities to transfer the theoretical knowledge from the classroom to a practical setting. Earlier works concurred with our findings and reported how students value blended learning, as it provides a platform for the application and discussion of newly acquired knowledge [22,53]. Therefore, future digital programs should not merely consist of lectures and discussions but a combination of interactive learning tools, such as practical exercises and quizzes, to engage students.

Nevertheless, students provided several improvements to further promote their engagement in the RISE program. With regards to the take-home tasks, students proposed to have more creative exercises. For instance, reflective practices may be embedded in other forms of activities as opposed to the traditional narrative form. Given that reflective practices are used in discussions [55] and in the form of photos [56], these activities can be considered as a form of reflective practice in future trials.

The findings from our qualitative study [28] proposed a resilience training program be held during the semester and right before major assessments. When enacted in this feasibility study, hosting the RISE program right before major assessments was found to be a challenge for students, especially for those with conflicting priorities. With more evidence showing that the timing of interventions affects learning outcomes [57], alternative initiation time points must be considered for future resilience trials.

### 4.1. Strengths and Limitations

This study has several strengths and limitations. First, this study adopted a theoretically driven, evidence-based, and learner-centered approach to develop the RISE program. Second, the adaptation of a mixed-methods design allowed a deeper exploration during the evaluation phase of the RISE program, which may not be achievable in a single design. Third, the findings from this feasibility study point toward the development of a larger-scale RCT to evaluate the effectiveness of the RISE program on improving students’ resilience, social support, and learning outcomes.

However, this study was limited by the use of a single-group pretest, post-test design, with the interpretation of its findings limited to the acceptability, appropriateness, and implementation of the RISE program. In addition, the study was only conducted in one university in Singapore. Considering the differences across institutions, the RISE program should be evaluated across multiple IHLs. Further, this study recruited a modest sample of 10 students as it sought to test the feasibility of the RISE program. Further work is needed to confirm the effects of the RISE program among undergraduate students. Finally, due to the COVID-19 pandemic, the use of open-ended questions to collect qualitative data through email correspondences may have limited participants’ responses. The use of in-depth virtual interviews should be considered in future trials to gain a deeper understanding of students’ experiences.

### 4.2. Implications for Future Research and Practice

This feasibility study lends its findings to the development of future resilience programs. First, larger-scale studies using RCT design are needed to confirm the effects of the RISE program in improving students’ resilience, social support, and learning outcomes. While resilience programs have been found to improve students’ resilience [16], there are limited studies evaluating the effectiveness of resilience training on social support and learning outcomes. The findings from this feasibility study present potential merits that warrant deeper exploration and evaluation.

Second, this study found that the students experienced competing priorities when the RISE program was delivered during the semester. Considering that students prefer to receive resilience training during the academic semester, future programs may be incorporated as part of the formal university curriculum. This may provide students with protected time to engage with the RISE program.

Third, the current RISE program adopts a blended learning design that may be challenging and resource-intensive. Future RCTs may consider comparing the effects of a blended and an asynchronous online resilience program. This has dual benefits, as it may reduce students’ workload and be less expensive to implement.

## 5. Conclusions

This study examined the feasibility of a digital RISE program. The findings show that students found the RISE program acceptable and appropriate and that they were able to engage with the materials. The limited efficacy testing also show that the digital RISE program can improve students’ resilience (*p* < 0.05) and meta-cognitive self-regulation (*p* < 0.05). However, due to the small sample size, it will be challenging to conclude the positive effects found in this study. Instead, this feasibility study paves the way for future studies using RCT designs to provide further evaluations of the RISE program on resilience, social support, and learning outcomes.

## Figures and Tables

**Figure 1 ijerph-19-12729-f001:**
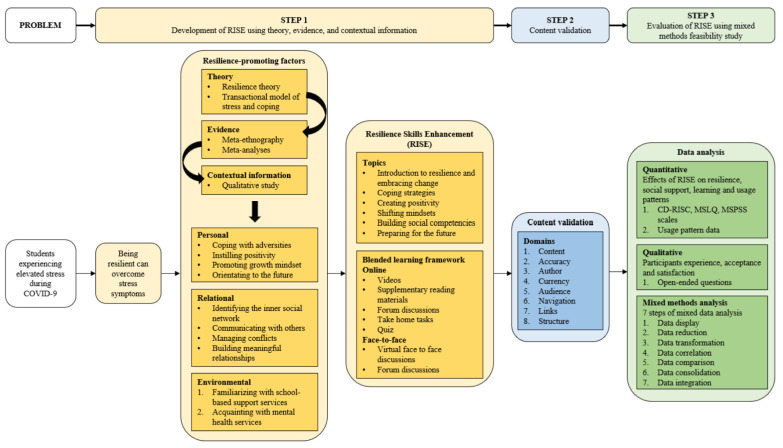
Overview of the development and evaluation process of RISE.

**Figure 2 ijerph-19-12729-f002:**
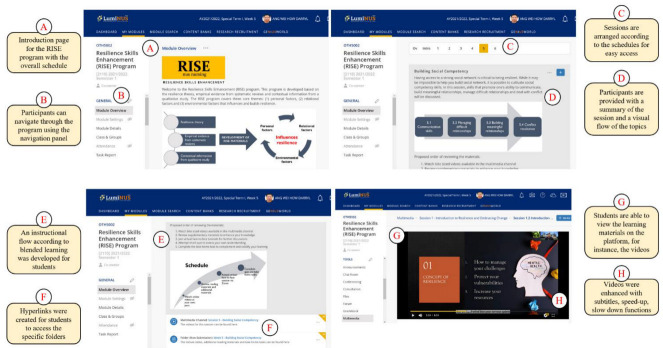
Web interface of the RISE program through the online learning platform.

**Figure 3 ijerph-19-12729-f003:**
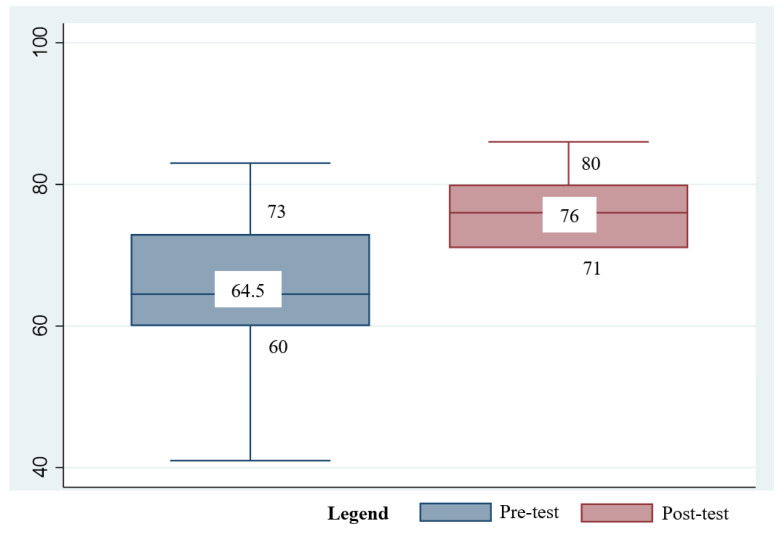
Box plot of resilience scores at the baseline and post-intervention.

**Table 1 ijerph-19-12729-t001:** Topics, session components, learning activities, and take-home tasks.

Session (Duration)	Aims	Rationale	Topics	Learning Activities			Learning Outcomes
				Preparatory	Online	FTF/Discussions	
Resilience and embracing change (78 min)	Present concept of resilience and readiness to incorporate resilience enhancing strategies	Contextualize resilience enhancing factors in the everyday life Initiate a behavioral change	Resilience and its promoting factors Embracing change	Reading materials Lecture notes	Videos Reflection Quiz	Virtual face-to-face via Zoom	Recognize the role of resilience in everyday life Appreciate the importance of behavioral changes to become resilience
Coping strategies (28 min)	Introduce personal, relational, and environmental coping strategies	Identify potential coping strategies from various sources	Coping strategies Resilient coping School-based academic services	Reading materials Lecture notes	Videos Practical exercise Quiz	Forum	Adopt suitable coping strategies to mitigate stress
Creating positivity (22 min)	Impart strategies to promote positivity	Appreciate the role of positivity Enhance emotion regulation	Introduction to positivity Strategies to create positivity	Reading materials Lecture notes	Videos Practical exercise Quiz	Forum	Ability to instill positivity and regulate emotions
Shifting mindsets (23 min)	Inculcating a growth mindset	Recognize the importance of growth mindset to overcome challenges	Fixed and growth mindsets Activating a growth mindset	Reading materials Lecture notes	Videos Practical exercise Quiz	Forum	Incorporate a growth mindset when experiencing any challenges
Building social competency (84 min)	Improve awareness of communication and social competencies skills	Develop meaningful relationships through effective communication	Communication techniques Managing relationships Conflict resolution	Reading materials	Videos Practical exercise Quiz	Virtual face-to-face via Zoom	Involve effective communication skills in managing relationships
Preparing for the future (128 min)	Provide insights towards career and finances	Obtain a future-orientated mindset Become aware of the available options after university	Financial management Job search and interviews Entrepreneurship	Reading materials	Videos Career-readiness exercise Quiz	Virtual face-to-face via Zoom	Implement strategies to work towards achieving one’s goals

FTF: Face-to-Face.

**Table 2 ijerph-19-12729-t002:** Sociodemographic and academic characteristics of the participants.

Characteristics		Number of Students	Percentage	Undergraduate Students ^#^	Proportions
**Age Mean = 21.5, SD = 1.96, Range = 19–24**					
**Gender**	Male	4	40%	15,528	49.6%
	Female	6	60%	15,744	50.4%
**Ethnicity**	Chinese	9	90%		74.1%
	Others	1	10%		3.2%
**Religion**	Buddhism	4	40%		33.3%
	Christianity	2	20%		18.3%
	Free thinker	2	20%		17%
	Islam	1	10%		14.7%
	Taoism	1	10%		10.9%
**Average household income**	<SGD2500 per person	3	30%	694,447 *	50.6%
	≥SGD2500 per person	7	70%	678,113 *	49.4%
**Type of residence**	<5-room HDB apartments	4	40%	765,479 *	55.8%
	≥5-room HDB apartments	6	60%	530,767 *	38.7%
**Financial aid**	Yes	3	30%		55%
	No	7	70%		45%
**Seniority**	Year 1	3	30%	8104	25.4%
	Year 2	1	10%	7276	23.4%
	Year 3	3	30%	7557	25.1%
	Year 4	3	30%	8230	26.1%
**Faculty**	Arts	2	20%	3295	10.5%
	Business	2	20%	4794	15.3%
	Science	3	30%	4584	14.7%
	Engineering	3	30%	4887	15.6%
	Nursing	2	20%	1120	3.6%
**Type of** **Program**	Single degree program	7	70%		
	Non-single degree program	3	30%		
**RISE duration ^1^**	Completion rate (%)	72	24.82		
	Total engagement (min)	182.66	92.85		

^#^: Data retrieved from 2021 NUS enrolment statistics. *: Data retrieved from 2020 Singapore’s census of population. ^1^: Mean and standard deviation.

**Table 3 ijerph-19-12729-t003:** Students’ acceptance, perceived appropriateness and demand, and actual use of the RISE program.

Categories	Sub-Categories	Quotations
Acceptance	Overall experience	“I really enjoyed the 6 weeks of the program, especially when it is held during the semester, the videos really motivate me and help me to find joy in what I’m doing in the midst of all the studying and hustling” (Year 2, Female, Nursing) “I am quite impressed by the curriculum and thought behind RISE. Very meaningful content and also discussed in a way that was informative and backed by frameworks and had theoretical underpinnings” (Year 4, Male, Arts) “Took some time for me to understand the flow, but after a few weeks, I got the hang of it and established a ‘routine’ I could use to go through the materials. But each of the materials is very user-friendly!” (Year 3, Female, Arts)
Perceived appropriateness	Content	“(the) RISE (program) gave insights as to how I cope with difficult situations and was a guide to understanding more about my own coping mechanisms so I could better know how to adjust to challenging circumstances in the future.” (Year 1, Male, Engineering)
	Take home tasks	“I actually like the reflection as it helps me to reflect back on my past experiences and think what could have been done better. This helps with my decision making in the future when dealing with challenges and changes.” (Year 3, Female, Nursing) “They [referring to the reflections] are alright, though it reminds me of the boring idea of essays as with other modules. This module [referring to RISE program] should have creative collaterals to peak interest among students.” (Year 4, Male, Arts)
	Duration	“The duration and training program is actually appropriate, though duration can be reduced as the module develops.” (Year 4, Male, Arts)
	Proposed improvements	“I would suggest including strategies on goal setting, focusing on the task at hand (avoiding distractions) and preventing procrastination.” (Year 2, Female, Nursing) “When to seek professional help and which types of help may be more suitable for specific situations.” (Year 1, Female, Arts)
Implementation	Facilitators	“I have been writing a gratitude journal ever since then, and I would say it has helped me to notice the little things that made me happy every day. I have also printed photos of the people that provide the greatest support to me and pasted it on my wall.” (Year 3, Female, Nursing) “I learned a lot more about myself such as my love language which could be a useful life skill in deciding the type of partner I go for.” (Year 1, Female, Arts)
	Barriers	“… the timing was a bit off since towards the tail-end of this program, we were all probably rushing for deadlines and submissions and final exams.” (Year 4, Male, Business) “The training is best conducted at the start of the term where there are fewer final year exams or projects.” (Year 4, Male, Business)

**Table 4 ijerph-19-12729-t004:** Effects of the RISE program on the students’ resilience, social support, and learning scores.

Outcome	Pretest (*n* = 10)				Post-Intervention (*n* = 9)				Z	*p*-Value ^#^	Effect Size (95% CI) ^1^	*p*-Value ^2^
	Median	IQR	Range	Mean (SD)	Median	IQR	Range	Mean SD)				
**Resilience**												
CD-RISC	64.5	60, 73	41–83	65.4 (11.62)	76	71, 80	48–98	75.78 (13.4)	−2.37	0.02 *	0.79 (−0.15, 1.74)	0.10
**Social support**												
MPSS	5.74	5, 5.92	2.17–7	5.24 (1.31)	6.17	5.67, 6.5	1.92–7	5.72 (1.51)	−1.67	0.09	0.33 (−0.58, 1.23)	0.48
MSPSS-FA	5.13	4.75, 5.25	2.5–7	4.95 (1.21)	6	5, 6.5	2–7	5.56 (1.55)	−1.54	0.14	0.42 (−0.49, 1.34)	0.36
MSPSS-FR	5.63	4.25, 7	3–7	5.33 (1.58)	6	5.75, 7	2.75–7	5.86 (1.35)	−0.81	0.43	0.34 (−0.57, 1.25)	0.46
MSPSS-SO	6	5, 6	1–7	5.45 (1.71)	6	6, 7	1–7	5.75 (1.85)	−1.98	0.13	0.16 (−0.74, 1.06)	0.73
**Motivated strategies for learning**												
MSLQ-CLB	5.13	4.5, 5.75	4–6.25	5.2 (0.73)	5.75	5.5, 6.25	4–7	5.81 (0.87)	−1.37	0.18	0.73 (−0.21, 1.67)	0.13
MSLQ-SELP	5.19	4.13, 5.5	2.7–6.6	5.03 (1.18)	5.75	5.13, 6.25	3.88–7	5.58 (1.02)	−1.68	0.11	0.47 (−0.44, 1.39)	0.31
MSLQ-TA	4.5	4.2, 5	2.4–6.6	4.36 (1.24)	4.4	2.2, 5.2	1.2–6.2	3.91 (1.74)	−1.36	0.20	−0.29 (−1.19. 0.62)	0.53
MSLQ-MCSR	4.5	4, 5.08	3.3–5.7	4.46 (0.76)	5.42	4.92, 5.67	4.8–6.5	5.40 (0.59)	2.43	0.01 *	1.31 (0.30, 2.33)	0.01 *
MSLQ-TSE	5.06	4.63, 5.75	4.4–6.4	5.16 (0.67)	5.63	5.38, 5.75	5.4–6	5.63 (0.23)	−1.49	0.14	0.88 (−0.08, 1.83)	0.07
MSLQ-ER	5.13	4.75, 5.75	4.5–6.3	5.25 (0.61)	5.5	5, 5.5	4.8–6.3	5.36 (0.44)	0.18	0.93	0.20 (−0.73, 1.12)	0.68

^#^: Wilcoxon signed-rank test. ^1^: Difference between the mean and standard deviation scores. ^2^: Effect size. *: *p* < 0.05. CD-RISC: Connor-Davidson Resilience Scale; MSPSS: Multidimensional Scale of Perceived Social Support; FA: Family; FR: Friends; SO: Significant Others; MSLQ: Motivated Strategies for Learning Questionnaire; CL: Control of Learning Beliefs; SELP: Self-efficacy of Learning and Performance; TA: Test Anxiety; MCSR: Meta-cognitive Self-Regulation; TSE: Time and Study Environment; ER: Effort Regulation.

## Data Availability

The data presented in this study are available on request from the corresponding author. The data are not publicly available due to ethical reasons.

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
