# Peer review of "Resilience for Undergraduate Students: Development and Evaluation of a Theory-Driven, Evidence-Based and Learner Centered Digital Resilience Skills Enhancement (RISE) Program"

_ijerph, 2022, doi:10.3390/ijerph191912729_

Round 1
Reviewer 1 Report
Thank you for the opportunity to read this interesting article. It is developed at a very good level.
The RISE program was also applied (in different forms) at other universities, in other countries. Therefore, it would be desirable to add appropriate comparisons of parameters achieved in similar studies in other countries to the Discussion section (comparison and discussion of results from Singapore and/versus other countries can enrich current science and practice even more).
A minor drawback is that only 10 students participated in the study (and even 1 did not even finish the survey).
Also, the Conclusion section could contain a summary of inspirations or recommendations that would help improve the quality of future RISE programs, not only in Singapore, but also in other countries.
Good luck!
Reviewer 2 Report
The manuscript is sound and concise. I have some remarks for the authors:
1) Remove the sentence in the abstract (or rephrase): 'Being resilient can iinouclate students against psychological distress and promote academic success'. This phrase does not add any information of the paper and the results do not match with the content of the phrase.
2) In the abstract you introduce a 'resilience skills enhancement program', which is not introduce. We need to know more information on the program, ... or rephrase the sentence.
3) Line 15, delete, were satisfied. Any rigorous analysis cannot be relied on satisfaction.
4) The term resilience is still too vague. Please consider, in the introduction, to provide with a clear definition of resilience, specially for enhancing students skills. Also, could you clearly link the resilience to the skills, by defining the significative skills, and which they are (and not some others).
5) Lines 127-134. They are very confusing in the terms of defining a program. Coul you state the characteristic educational activities that nbest promote students' resilience?
6) The 2.3.3: It seems that part of the paragraph should be considered as results.
7) Figure 2 is not ina good format. I couldn't read it. And this is a major problem.
8) You should also link resilience to the control of learning, self-efficacy and anxiety. To me, the term learning beliefs is too much confusing. Why motivation is related to anxiety?
9) In lines 243 to 250, the interreliability between analysers, should be described.
10) At the very end, the manuscript reports that the study is about a learner-centered digitial resilience. Then, why not defining the 'digitla resilience' from the very begining of the manuscript?
11) lines 376 to 388 are not necessary in a discussion section.
12) Between line 389 and 395, you use the terms, rigorous, elaborated, important, great value, ... This is a bad use of the narrative, it is jargon, which should be avoided.
13) The discussion is quite superficial, since the use of the references is too generalist and not adding value. Please, be more critical. You shoul dcompare the results of the application of the program and the 'significtive' categories with studies that also analyse for example, the learning outcomes, or the resilience, etc.
14) The conclusions should avoid the 'acceptable and appropriate' considerations. You need to provide scientific results.
Reviewer 3 Report
Thanks for submitting this manuscript for review. It was a pleasure reading this paper. However, I do have some questions.
1. In terms of the instruments, one of the outcome measure is MSLQ? Could you please explain why did you use the original version MSLQ? This instrument had been adapted and validated in many other context that might be more close to Singapore, why not choose a more updated version?
2. The sample size of the current study is very limited. However, based on the description of the program, which indicates all undergraduate full-time students can access, it rise concerns that only 10 students were involved in the current study.
3. RISE program has a complicated design with various measures, however, based on the current analysis, it is hard to generate the conclusion that RISE is an acceptable and appropriate program.
4. MSLQ was used in the study to measure motivation and self-regulated learning. However, it was not indicated in the previous sections why this factor is related to the program.
Round 2
Reviewer 2 Report
Dear authors,
the mansucript has gain in clarity. The manuscript is now sound and concise.
All my best
Reviewer 3 Report
All the comments are well addressed. Thank you!